# Influence of Clinical Factors on the Quality of Life in Romanian People with Epilepsy—A Follow-Up Study in Real-Life Clinical Practice

**DOI:** 10.3390/jpm13050752

**Published:** 2023-04-28

**Authors:** Ionut-Horia Cioriceanu, Dan-Alexandru Constantin, Elena Bobescu, Luigi Geo Marceanu, Liliana Rogozea

**Affiliations:** 1Clinical Hospital of Psychiatry and Neurology Brasov, 500123 Brasov, Romania; dr.cioriceanu@gmail.com; 2Department of Fundamental, Prophylactic and Clinical Sciences, Faculty of Medicine, Transylvania University of Brasov, 500019 Brasov, Romania; 3Department of Medical and Surgical Specialties, Faculty of Medicine, Transilvania University of Brasov, 500019 Brasov, Romania

**Keywords:** epilepsy, quality of life, QOLIE-31-P

## Abstract

Background: This study aimed to assess the influence of various clinical factors on the quality of life perception of patients with epilepsy over a follow-up period in current clinical practice. Methods: Thirty-five PWE evaluated via video-electro-encephalography in the Clinical Hospital of Psychiatry and Neurology in Brasov, Romania, were included, and the quality of life was assessed using the Romanian version of the QOLIE-31-P questionnaire. Results: At baseline, the mean age was 40.03 (±14.63) years; the mean duration of epilepsy was 11.46 (±12.90) years; the mean age at the first seizure was 28.57 (±18.72); and the mean duration between evaluations was 23.46 (±7.54) months. The mean (SD) QOLIE-31-P total score at the initial visit (68.54 ±15.89) was lower than the mean (SD) QOLIE-31-P total score at the follow-up (74.15 ± 17.09). Patients with epileptiform activity recorded via video-electro-encephalography, using polytherapy, those with uncontrolled seizures, and those with one or more seizures per month had statistically significantly lower QOLIE-31-P total scores at baseline and follow-up. Multiple linear regression analyses revealed seizure frequency as a significant inverse predictor of quality of life in both evaluations. Conclusions: The QOLIE-31-P total score was improved during the follow-up period, and medical professionals should use instruments to evaluate quality of life and identify patterns while trying to improve the outcomes of patients with epilepsy.

## 1. Introduction

Epilepsy is a chronic neurological condition identified by the recurrence of seizures. It is an important health problem that affects 1–2% of the global population and can have a significant impact on the quality of individual life [1]. If patients with epilepsy (PWE) are well-controlled, their perception of the quality of life (QOL) is similar to that of healthy people; if this does not happen, their perception is worse than that of the general population or comparable to or worse than that of patients with other chronic diseases [2]. The level of QOL is negatively correlated with the frequency of seizures in most studies [3,4,5,6]. There are numerous clinical factors reported by other authors that have an impact on the QOL, such as the duration of the disease [7], the age of onset [8], the type of seizures [9] or the number of anti-seizure medications (ASM) [10]; however, the determining role of these variables differs between countries.

Despite the fact that there are significant studies on the QOL of PWE, longitudinal studies are rare in the international scientific literature and non-existent in Romania in recent years.

Through an outpatient follow-up period, it is important to understand how PWE perceive their QOL. By studying the factors that influence perception, both the research strategies and the management of these patients can be improved.

The aim of this study was to evaluate the QOL perception of PWE and the influence of different clinical factors over a follow-up period in real-life clinical practice.

## 2. Materials and Methods

### 2.1. Sample

Our initial study [11] included patients diagnosed with epilepsy according to the International League Against Epilepsy (ILAE) criteria [12] referred for video-electro-encephalography (VEEG) evaluation at the Clinical Hospital of Psychiatry and Neurology Brasov between February 2018 and August 2021. They were invited, and then provided with details of the study’s objective. They were offered the possibility to ask questions about the research, and it was explicitly stated to them that declining to participate would not put them through any inconvenience. Patients were enrolled if they consented to participate. Only individuals with another progressing neurological disease, mental disorder, intellectual impairment, or those who had issues understanding were excluded; no specific patients were selected.

From the 91 patients in the initial study, we were able to retest and perform a follow-up study of the QOL of 35 PWE. Patients who did not reschedule themselves for medical appointments were invited via phone call, and data collection has been completed at the Video-EEG Unit on the date scheduled between August 2020 and August 2021. The duration between assessments was influenced by the availability of PWE and restrictions associated with the COVID-19 pandemic. Those who could not be retested provided the following reasons: (1) they changed their residence to another city and could not travel the distance; (2) they feared infection with the SARS-CoV-2 virus and the restrictions; (3) they could not be contacted because they changed their phone number or did not respond to the email; and (4) they did not want an appointment at the moment.

This study was approved by the Research Ethics Commission of the Transylvania University of Brasov. Before enrolling, written informed consent was signed by the patients who agreed to take part in the study for the publication of the findings.

### 2.2. Instruments

The Patient-Weighted Quality of Life in Epilepsy Inventory—QOLIE-31-P© [13], an epilepsy-specific measure of QOL designed for use in adults 18 years and older—was used. It is composed of 38 items evaluating 7 domains: energy, mood (emotional functioning), daily activities (social functioning), cognition (cognitive functioning), medication effects, seizure worry and overall QOL. Each domain scale has one item that addresses the level of distress for the respondent related to worries associated with the disease that is used as weighting. The scoring process corresponds with that of the scoring manual for QOLIE-31-P, and the scores range from 0 to 100, with higher values being indicative of better QOL [14]. Before applying the inventory, we asked for permission. Cramer J. provided and returned the Romanian version of the QOLIE-31-P that we used in this research. The QOLIE-31-P was previously used in Romanian PWE clinical trials [15,16].

The following data were collected: (1) sociodemographic data: age, sex, last level of completed studies, socio-professional category, living environment, marital status; (2) clinical data about epilepsy: age of onset, type of seizures, etiology, type of epilepsy by onset, presence of aura, frequency of seizures, presence of seizures in sleep and number of ASM; and (3) information about the location of the epileptiform activity based on VEEG recordings that were performed on each participant. The classification of the seizures was carried out taking into account several criteria: (1) according to the frequency of seizures reported by the patient at the initial and final assessments, three groups were described: those with one or more seizures per month, those with one to six seizures per year, and those with no seizures in the last year; (2) depending on the last episode of seizures, the patients were included in two groups: patients with uncontrolled epilepsy, those who had at least one seizure in the last year; and patients with controlled epilepsy, those who did not have seizures in the last year. It was noted at the follow-up if there was a difference in the frequency of seizures when compared to the initial evaluation.

### 2.3. Statistical Analysis

GraphPad Prism version 9.2.0 (San Diego California) was used for statistical analysis, considering for all tests that the results are statistically significant at a value of *p* less than 0.05; the confidence interval is 95%. The analysis was performed dynamically, taking into account the results obtained at the initial and final visits, studying the evolution of the QOL perception of PWE and the relationship between the QOLIE-31-P data and the socio-demographic data, respectively, and the clinical variables of epilepsy.

Quantitative variables were described as means and standard deviation (SD) and categorical variables using absolute values and percentages. For testing the associations, Chi-square test was used for the categorical variables; independent Student’s *t*-test was used for two groups; and one-way analysis of variance was used for >2 groups, which were used for the quantitative variables. To test for the predictors of the QOLIE-31-P TS, significant variables found in the univariate analysis were entered into a multiple linear regression model.

## 3. Results

A total of 35 PWE were studied in the initial and final evaluations. The mean (SD) duration between evaluations was 23.46 (±7.54) months. In the initial assessment, PWE had a mean (SD) age of 40.03 (±14.63) years, aged 28.57 (±18.72) at first seizure, and had a mean duration of epilepsy of 11.46 (±12.90) years. A total of 60% (*n* = 21) were females, 80% (*n* = 28) lived in urban regions, 54.28% (*n* = 19) were married, and 34.28 % (*n* = 12) had a high formal level of education. According to ILAE criteria, structural epilepsy was noted in 57.14% (*n* = 20) of cases, unknown in 40% (*n* = 14) of cases and genetic in 2.85% (*n* = 1) of cases. The majority of the patients had bilateral tonic-clonic seizures, without aura, without seizures in their sleep and had epileptiform activity during prolonged video-EEG recordings. By onset, focal seizures were noted in 94.28% (*n* = 33) and generalized in 5.71% (*n* = 2) of cases. At the follow-up visit, 60% (*n* = 21) of patients were controlled, compared with 14.21% (*n* = 5) at the initial evaluation, and 20% (*n* = 7) of patients experienced at least one seizure per month compared with 34.28% (*n* = 12) at the initial evaluation. The sociodemographic and clinical aspects of epilepsy and QOLIE-31-P TS in the initial and final evaluations are shown in Table 1.

The mean QOLIE-31-P TS at the initial visit (68.54 ±15.89) was lower than the mean QOLIE-31-P TS at the follow-up visit (74.15 ± 17.09). On the domains’ scores, at the initial visit, the highest mean score was for cognition (51.71 ± 35.40), while the lowest mean score was for seizure worry (32.43 ± 31.28). At the final evaluation, the highest mean score was for cognition (66.39 ±32.92), while the lowest mean score was for energy (45.24 ± 31.58). The highest mean value difference between the final and initial evaluation was found for the medication effects domain. QOLIE-31-P domain scores in the initial and final evaluation and the difference between the mean values are shown in Table 2.

Comparing the groups, both at the initial and final evaluation, patients with epileptiform activity recorded via VEEG who were taking two or more ASM, those with uncontrolled seizures, and those with one or more seizures per month had statistically significantly lower QOLIE-31-P TS. In the final evaluation, the presence of aura was associated with a lower QOLIE-TS, comparing with those who had no signs before seizures. After multiple linear regression analyses, though the initial evaluation number of ASM taken was associated with QOLIE-31-P TS, only seizure frequency remained significantly associated in both evaluations and had the most important influence on the QOLIE-31-P TS (Table 3).

## 4. Discussion

Considering the unpredictability of the COVID-19 pandemic, we were able to carry out this study and observe that at follow-up, the QOLIE-31-P TS and domain scores of PWE, which were evaluated at baseline, were higher. This may reflect that despite the consequences of COVID-19, including economic difficulties and challenges in healthcare systems, the perception of our studied patients improved compared with recently published studies focusing on the QOL of PWE in the era of COVID-19 [17,18]. In the literature, there are few longitudinal studies before the pandemic, none performed in Romania, and most of them are clinical trials with new ASM or with patients with temporal lobe epilepsy undergoing surgery.

At baseline and follow-up, higher seizure frequency predicted lower QOLIE-31-P TS in accordance with other studies where high frequency of seizures was a significant predictor of poorer QOL [4,7,9,19]. At follow-up, the number of controlled patients with no seizures in the last year was higher, and the better control of seizures was correlated with an increase in QOLIE-31-P TS and improved perception. Studies that investigated the evolutionary characteristics of the QOL perception in patients who changed ASM or had undergone epilepsy surgery showed an improvement in QOL in PWE with a decrease in the frequency of seizures and particularly in those who became seizure-free [20,21,22]. Our results provide additional evidence that efforts to lower seizure frequency are important for enhancing quality of life.

At both evaluations, patients in our study using two or more ASM had a worse QOL perception compared to those using monotherapy. In multivariate analysis, only at initial evaluation did the number of AEM taken remain significantly associated with QOLIE-31-P TS. Our findings are partially in line with earlier research that has described the influence of polytherapy on QOL [23,24]. Other studies have found no correlations between the numbers of ASM and QOL [25,26]. The impact of medication on QOL may be correlated with the probability of adverse effects, confidence in treatment and compliance. Future studies are needed to evaluate the impact of the number of ASM used in PWE on the QOL of these patients in Romania.

Patients in this study were evaluated via VEEG and the presence of epileptiform activity was studied as part of the management of diagnosis and treatment. We found that patients with abnormal EEG signals during VEEG monitoring had lower QOLIE-31-P TS at baseline and follow-up. We did not find any statistically significant association between QOLIE-31-P-TS and age of onset, seizure type, presence of seizures in sleep, epilepsy type, etiology or socio-demographic characteristics in the initial and final evaluations. These factors have already been described in different research and cultures [5,11,27], and continuing to study predictors of QOL may enhance clinical practice and support for PWE.

Our study had some limitations. Held during a period that also included the period of the COVID-19 pandemic, in which the access to non-COVID patients was limited both by the restrictions imposed by the medical authorities and by the fear shown by PWE to present themselves in a medical care unit and potentially contract an infection, the number of participants in the study was smaller, which can lead to difficulties in statistical analysis. Another limitation is related to the fact that the study was conducted in a single institution; however, this is offset by the rigor with which it was conducted and the use of the same evaluation standards. From the beginning, the COVID-19 pandemic in Romania had a similar pattern to that in other European countries, with our country going through epidemiological countermeasures including total lockdown, followed later by gradual partial openings [28]. The hospital where the research took place was listed among the national hospitals that treated patients infected with the SARS-CoV-2 virus as of April 2020. The new pandemic produced a series of limitations in the ability to provide medical assistance related to the activities of medical staff but also to the medical and public health intervention [29,30]. These limitations include, in the case of PWE, the inability to use the only VEEG unit in the county, located in one of the sections of the hospital that has become a support health unit for the treatment of patients with COVID-19. Under these conditions, the PWE evaluations were postponed for a period of time, and a series of important ethical decisions had to be made, including those related to the accessibility of health care [31]. Because of the small number of reevaluated patients and the unpredictability of the pandemic, we were unable to conduct a more extensive evaluation regarding the impact of the restrictions or infection with SARS-COV-2 on the QOL. According to earlier research that showed that stress levels are high in the case of PWE [32], the probability of contact with COVID-19 patients and a potential diagnosis of COVID-19 can justify the refusal of certain patients to present themselves for re-evaluation. An international agreement on PWE determined that PWE should avoid hospitals where there is a risk of COVID-19 spreading, continue their treatment at home, and be given access to regular ASM [33]. No screening for depressive symptoms was carried out because we only do so when we suspect PWE of having these and seek additional consultation with a psychologist.

## 5. Conclusions

This follow-up study in real-life clinical practice in a Romanian public hospital has shown that QOLIE-31-P TS is lower in PWE with frequent seizures, uncontrolled disease, those using polytherapy, and those with epileptiform activity recorded on VEEG.

Seizure frequency was a significant inverse predictor of QOL and because epilepsy is a chronic condition that requires regular outpatient visits, medical professionals should use instruments to evaluate QOL, identify patterns, and take into account various clinical factors while trying to improve PWE outcomes. Future studies should include an increased number of participants from all over the country to facilitate comparisons between different patient groups or populations and consider the influence of psychological profiles for more accurate results.

## Figures and Tables

**Table 1 jpm-13-00752-t001:** Sociodemographic and clinical aspects of epilepsy and QOLIE-31-P TS in the initial and final evaluations.

		Total Score	*p*
Initial evaluation			
Age	18–44 (*n* = 23)/≥45 (*n* = 12)	69.54 (±14.79) vs. 66.61 (±18.35)	0.6119
Sex	Female (*n* = 21)/Male (*n* = 14)	68.18 (±15.50) vs. 69.08 (±17.02)	0.8724
Environment	Urban (*n* = 28)/Rural (*n* = 7)	68.74 (±16.26) vs. 67.74 (±15.47)	0.8842
Marital Status	Non-married (*n* = 16)/Married (*n* = 19)	70.01 (±15.07) vs. 67.29 (±16.85)	0.6211
Employment status	Employed (*n* = 19)/Unemployed (*n* = 16)	71.09 (±14.73) vs. 65.51 (±17.14)	0.3077
Education	University (*n* = 12)/Other (*n* = 23)	71.46 (±13.81) vs. 67.01 (±16.96)	0.4398
Age of onset (years)	<18 (*n* = 12)/>18 (*n* = 23)	67.78 (±15.74) vs. 68.93 (±16.30)	0.8424
Seizure type	With (*n* = 32)/Without motor tonic-clonic (*n* = 3)	69.50 (±15.84) vs. 58.22 (±15.01)	0.2452
Seizures in sleep	Yes (*n* = 6)/No (*n* = 29)	77.72 (±12.88) vs. 66.64 (±15.97)	0.1215
Epilepsy type (onset)	Focal (*n* = 33)/Generalized (*n* = 2)	68.17 (±16.02) vs. 74.59 (±16.91)	0.5865
Etiology	Unknown (*n* = 14)/Structural (*n* = 20)/Genetic (*n* = 1)	69.16 (±18.64) vs. 68.40 (±14.56) vs. 63.63 (±0.00)	0.9268
Presence of aura	Yes (*n* = 16)/No (*n* = 19)	64.16 (±15.78) vs. 72.22 (±15.43)	0.1371
Epileptiform activity	With (*n* = 32)/Missing (*n* = 3)	67.59 (±16.31) vs. 78.61 (±1.160)	**0.0007**
Seizure control	Controlled (*n* = 5)/Uncontrolled (*n* = 30)	77.27 (±5.187) vs. 67.08 (±16.64)	**0.0145**
Seizure frequency	One or more seizures per month (*n* = 12)/other (*n* = 23)	54.20 (±14.53) vs. 76.02 (±10.67)	**<0.0001**
Number of ASM taken	One (*n* = 16)/≥2 (*n* = 14)/Without (*n* = 5)	75.39 (±12.29) vs. 55.99 (±13.63) vs. 81.72 (±4.385)	**<0.0001**
Final evaluation			
Age	18–44 (*n* = 19)/≥45 (*n* = 16)	77.43 (±14.22) vs. 70.26 (±19.73)	0.2212
Presence of aura	Yes (*n* = 16)/No (*n* = 19)	66.20 (±17.28) vs. 80.85 (14.12)	**0.0093**
Epileptiform activity	With (*n* = 32)/Missing (*n* = 3)	73.23 (±17.55) vs. 84.00 (±5.446)	**0.0433**
Seizure control	Controlled (*n* = 21)/Uncontrolled (*n* = 14)	81.91 (±12.13) vs. 62.52 (±17.16)	**0.0004**
Seizure frequency	One or more seizures per month (*n* = 7)/Other (*n* = 28)	51.31 (±17.48) vs. 79.86 (±11.44)	**<0.0001**
Number of ASM taken	One (*n* = 18)/≥2 (*n* = 15)/Without (*n* = 2)	79.94 (±9.575) vs. 62.13 (±19.12) vs. 89.30 (±4.767)	**0.0001**

**Table 2 jpm-13-00752-t002:** QOLIE-31-P domain scores in the initial and final evaluation.

	Initial	Final	Difference Mean Values
Energy	37.41 (±28.78)	45.24 (±31.58)	7.83
Mood	38.33 (±25.49)	47.53 (±29.86	9.20
Daily Activities	49.17 (±35.13)	56.08 (±34.67)	6.91
Cognition	51.71 (±35.40)	66.39 (±32.92)	14.68
Medication Effects	44.97 (±30.96)	60.40 (±35.75)	15.43
Seizure Worry	32.43 (±31.28)	47.46 (±35.11)	15.03
Overall Quality of Life	42.34 (±25.47)	55.31 (±27.71)	12.97

**Table 3 jpm-13-00752-t003:** Initial and final evaluation variables associated with QOLIE-31-P TS in linear regression.

	Estimate	Standard Error	95% CI	|t|	*p* Value
Parameter estimates					
Initial					
Seizure frequency	−17.11	5.568	−28.60 to −5.619	3.073	0.0052
Number of ASM taken	−15.95	7.139	−30.69 to −1.220	2.235	0.0350
Final					
Seizure frequency	−18.94	7.939	−35.32 to −2.550	2.385	0.0253
Number of ASM taken	−13.73	10.69	−35.80 to 8.344	1.284	0.2115

## Data Availability

The data presented in this study are available on request from the corresponding author.

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
