# Peer review of "Influence of Clinical Factors on the Quality of Life in Romanian People with Epilepsy—A Follow-Up Study in Real-Life Clinical Practice"

_jpm, 2023, doi:10.3390/jpm13050752_

Round 1

Reviewer 1 Report

The article evaluated different clinical factors on quality of life in PWE with the Romanian version of the QOLIE-31-P questionnaire. The study demonstrated that patients with epileptiform activity, the use of polytherapy and uncontrolled seizures are factors that reduce the quality of life

Comments

I suggest in the introduction inserting information on clinical factors that affect quality of life in other studies.

Insert in the introduction the meaning of PWE,QOL,VEEG

In the discussion, the authors could consider stress as a clinical factor, this because the study was carried out during COVID, a critical period of fear, stress and vulnerability to infection.

Author Response

The article evaluated different clinical factors on quality of life in PWE with the Romanian version of the QOLIE-31-P questionnaire. The study demonstrated that patients with epileptiform activity, the use of polytherapy and uncontrolled seizures are factors that reduce the quality of life

Comments

I suggest in the introduction inserting information on clinical factors that affect quality of life in other studies.

Revised – We consider that we enumerated in the introduction clinical factors such as frequency of seizures, duration of the disease, age of onset, type of seizures, or number of antiseizure medications. In the discussions, the data regarding the clinical factors are presented in more detail and compared with the data from the literature.

The level of QOL is negatively correlated with the frequency of seizures in most studies [3–6]. There are numerous clinical factors reported by other authors that have an impact on the QOL, such as the duration of the disease [7], the age of onset [8], the type of seizures [9] or the number of anti-seizure medications (ASM) [10], but the determining role of these variables differs between countries.

Insert in the introduction the meaning of PWE,QOL,VEEG

Changed – inserted in the introduction

In the discussion, the authors could consider stress as a clinical factor, this because the study was carried out during COVID, a critical period of fear, stress and vulnerability to infection.

Changed – added “According to earlier research that showed that stress levels are high in the case of PWE [32] , the probability of contact with COVID-19 patients and a potential diagnosis of COVID-19 can justify the refusal of certain patients to present themselves for re-evaluation. An international agreement on PWE determined that PWE should avoid hospitals where there is a risk of COVID-19 spreading, continue their treatment at home, and be given access to regular ASM [33]”

Reviewer 2 Report

In this manuscript, authors evaluated various clinical factors on the quality of life (QOL) perception of patients with epilepsy (PWE) over a follow-up period. They chose the Patient-Weighted Quality of Life in Epilepsy Inventory, QOLIE-31-P, an epilepsy-specific measure of QOL designed for use in adults 18 years and older. A total of 35 PWE were studied in the initial and final evaluations. They found that the total score of QOLIE-31-P was improved during follow-up period, and provide additional evidence that efforts to lower seizure frequency are important for enhancing quality of life. This manuscript is well written overall. I have comments listed below.

1. There are 91 patients in the initial study, and the authors were able to retest and perform a follow-up study of the QOL of 35 PWE. The number of reevaluated patients is small due to various reasons. However, the data of the remaining 56 patients in the initial evaluation is still important. Therefore, I wonder if the authors are able to provide these data. Even though these data are not comparable to the follow-up evaluation, the increased patient number may lead to more reliable conclusions in terms of initial evaluation.

2. Does the duration between evaluations also have a contribution to the total score of the final evaluations? Since the mean (SD) duration between evaluations was 23.46 (±7.54) months, I’m not sure if the authors could analyze among the cohort. Anyway, it’s worthy to discuss a bit more.

3. Could the authors generate a graphical plot to present the data in Table 2. That would help readers understand the point of this study.

4. Table 1, third row: Age 18-44 (n=23) / 44 (n=12): Are there 12 patients with age 44? I believe there is a mistake since it does not match the age in the final evaluation.

Author Response

In this manuscript, authors evaluated various clinical factors on the quality of life (QOL) perception of patients with epilepsy (PWE) over a follow-up period. They chose the Patient-Weighted Quality of Life in Epilepsy Inventory, QOLIE-31-P, an epilepsy-specific measure of QOL designed for use in adults 18 years and older. A total of 35 PWE were studied in the initial and final evaluations. They found that the total score of QOLIE-31-P was improved during follow-up period, and provide additional evidence that efforts to lower seizure frequency are important for enhancing quality of life. This manuscript is well written overall. I have comments listed below. 

  1. There are 91 patients in the initial study, and the authors were able to retest and perform a follow-up study of the QOL of 35 PWE. The number of reevaluated patients is small due to various reasons. However, the data of the remaining 56 patients in the initial evaluation is still important. Therefore, I wonder if the authors are able to provide these data. Even though these data are not comparable to the follow-up evaluation, the increased patient number may lead to more reliable conclusions in terms of initial evaluation.

Revised - The data for the initial 91 patients were presented in our other study published last year, which is cited in the text. Therefore, the data of the other patients is available, but they were not part of the design of this study.

  1. Does the duration between evaluations also have a contribution to the total score of the final evaluations? Since the mean (SD) duration between evaluations was 23.46 (±7.54) months, I’m not sure if the authors could analyze among the cohort. Anyway, it’s worthy to discuss a bit more. 

Revised – We did not investigate the impact of the period between evaluations on the total score because it was not part of the design of the study. We did not establish a specific period; the study was carried out in current practice. At initial evaluation, we instructed patients, as we usually do in Romania, to come to check-up whenever they subjectively considered that there were changes in their condition status. The patients showed up for the reevaluation when they made their own appointment. Later, depending on the lifting of the restrictions, those who did not present themselves were called and invited. The outcome of the influence of the duration between assessments will be studied in a future study that is in the enrollment phase.

  1. Could the authors generate a graphical plot to present the data in Table 2. That would help readers understand the point of this study. 

Revised – This article is the naturally continuation of the previous article and we wanted to keep the design line and graphics for a better understanding and comparison.

  1. Table 1, third row: Age 18-44 (n=23) / 44 (n=12): Are there 12 patients with age 44? I believe there is a mistake since it does not match the age in the final evaluation.

Changed - Indeed, there was a typo in the table that we corrected – “Age 18-44 (n=23) / ≥45 (n=12)”